# Optimising risk-based surveillance for early detection of invasive plant pathogens

Alexander J. Mastin[1]*, Timothy R. Gottwald[2], Frank van den Bosch[1,3], Nik J. Cunniffe[4‡], Stephen Parnell[1‡]

1 Ecosystems and Environment Research Centre, School of Science, Engineering and Environment, University of Salford, Greater Manchester, United Kingdom, 2 USDA Agricultural Research Service, Fort Pierce, Florida, United States of America, 3 Department of Environment and Agriculture, Centre for Crop and Disease Management, Curtin University, Bentley, Perth, Australia, 4 Department of Plant Sciences, Downing Street, Cambridge, United Kingdom

‡ These authors are joint senior authors on this work.
* a.mastin@salford.ac.uk

**Data Availability Statement:** Due to commercial sensitivities, high resolution citrus density data are not available. We have provided lower resolution data, along with the model code, in the GitHub repository (https://github.com/nikcunniffe/

## Abstract

Emerging infectious diseases (EIDs) of plants continue to devastate ecosystems and livelihoods worldwide. Effective management requires surveillance to detect epidemics at an early stage. However, despite the increasing use of risk-based surveillance programs in plant health, it remains unclear how best to target surveillance resources to achieve this. We combine a spatially explicit model of pathogen entry and spread with a statistical model of detection and use a stochastic optimisation routine to identify which arrangement of surveillance sites maximises the probability of detecting an invading epidemic. Our approach reveals that it is not always optimal to target the highest-risk sites and that the optimal strategy differs depending on not only patterns of pathogen entry and spread but also the choice of detection method. That is, we find that spatial correlation in risk can make it suboptimal to focus solely on the highest-risk sites, meaning that it is best to avoid 'putting all your eggs in one basket'. However, this depends on an interplay with other factors, such as the sensitivity of available detection methods. Using the economically important arboreal disease huanglongbing (HLB), we demonstrate how our approach leads to a significant performance gain and cost saving in comparison with conventional methods to targeted surveillance.

## Introduction

The collapse of the American chestnut population in the eastern United States in the early 20th century [1], the English elm in 1960s and 1970s UK [2], tan oak and coast live oak in the western United States over the last 30 years [3], and the citrus industry in Florida since 2005 [4] have all resulted from the emergence of pathogens that were not previously present. Emerging infectious diseases (EIDs) such as these are an increasing threat to wild and cultivated plants worldwide [5–7]. In some cases, EIDs may be established pathogens that have moved into new areas, as exemplified by the ongoing spread of the bacterium causing the citrus disease huanglongbing (HLB) in the US [8] and a competent vector of this pathogen in Europe

SpatialSampling). All data required to create the figures in the manuscript (with the exception of the high resolution citrus density data in Fig 1A) are available in the figshare repository (https://salford.figshare.com/account/articles/12759929; DOI 10.17866/rd.salford.12759929).

**Funding:** SP was partly funded by the USDA-APHIS farm bill grant (17-8130-0570-CA). https://www.fsa.usda.gov/programs-and-services/farm-bill/index SP was partly funded by Defra. https://www.gov.uk/government/organisations/department-for-environment-food-rural-affairs FvdB received funding from the BBSRC whilst at Rothamsted Research. https://bbsrc.ukri.org/ The funders had no role in study design, data collection and analysis, decision to publish, or preparation of the manuscript.

**Competing interests:** The authors have declared that no competing interests exist.

**Abbreviations:** EID, emerging infectious disease; HLB, huanglongbing; Las, *Candidatus* Liberibacter asiaticus; USDA-APHIS, United States Department of Agriculture Animal and Plant Health Inspection Service; USDA-ARS, United States Department of Agriculture Agricultural Research Service.

[9]. In other cases, completely new pathogen strains may emerge, as seen with the emergence of the CoDiRO strain of *Xylella fastidiosa* (*X. fastidiosa* subsp. pauca, sequence type [ST]53: the cause of olive quick decline syndrome) in Italy [10] and new strains of *Puccinia graminis* (the cause of wheat stem rust) worldwide [11]. Although changes in farming practices, land use, and climate are important drivers of these processes [6], most attention to date has focused on the global movement of people, plants, and plant products through travel and trade [12,13]. However, efforts to target these pathways directly through movement restrictions, quarantine, and border inspection [14] often fail [15], allowing pathogens to enter and potentially establish and spread to levels at which control efforts are infeasible [16]. As a result, it is increasingly recognised that surveillance activities must already be in place in the region of interest before pathogen entry [17] and must be capable of detection before the prevalence (that is, the proportion of the host population that is infected) exceeds the point at which control is almost certain to be no longer possible. This 'maximum acceptable prevalence' will be impacted by factors such as the epidemiology of the pest, its likely impact, and the availability and feasibility of control measures.

Developing an effective early detection surveillance strategy is complicated by the need to survey large, heterogeneous, areas of landscape over an indefinite timespan in the face of limited financial and logistical resources. Whilst it is well accepted that attention must be focused on sampling enough hosts regularly enough for the likely prevalence at first detection to be acceptably low [18], consideration must also be given to which hosts should be inspected and where to sample. One way of achieving this is through 'risk-based' or 'targeted' surveillance [19,20], in which the types of hosts or locations judged most likely to contain the pest or pathogen are preferentially selected for inspection or sampling [21]. Although the merits of targeted surveillance are well recognised, most work to date has focused on identifying static 'high-risk' groups and locations using statistical models [22,23]. Although these methods of planning targeted surveillance are a valuable and versatile method of quantifying the infection risk amongst different groups, they do not explicitly account for the epidemiological processes that determine where and, importantly, when a pathogen will be present. As a result, there is a risk that the surveillance strategy may not be optimally targeted, resulting in low performance and/or excessive costs—both of which can ultimately lead to surveillance system failure.

The inability of conventional risk-based strategies to account explicitly for the epidemiology of a pest or pathogen also has important implications for early detection surveillance planning. To take a simple example, a common targeted surveillance strategy for EIDs is to focus on areas where the pathogen is more likely to first enter [24]. However, conventional methods do not tell us whether resources should all be placed around the single highest-risk site or spread across other potential introduction sites as well. Such questions can be answered by considering the placement of surveillance resources as an optimisation problem [25]. By linking spatial and/or temporal simulation models that replicate the spread of the pest or pathogen to computational optimisation routines to identify particular sampling patterns, precise surveillance and/or control strategies that minimise the impact of the pest or pathogen can be identified [26]. Although much work to date has focused on identifying how best to conduct surveys in order to achieve certain disease management or mitigation objectives whilst considering spatial spread of a pest or pathogen [27–32] and on the value of different network metrics for identifying hosts to target for surveillance [33–37], there has been little work on the optimal deployment of surveillance resources for early pest or pathogen detection that explicitly considers the spread of the agent through a real-world landscape. No previous study has addressed the pivotal question: where exactly should surveillance resources be located to maximise the probability to detect an invading pathogen before it reaches a certain prevalence?

We propose and test a novel, to our knowledge, approach to surveillance planning that explicitly accounts for the spatial spread of a pathogen and that can be readily and easily applied to any pest or pathogen of interest. Using a spatially explicit, stochastic, epidemiological model, we represent the processes of pathogen introduction and onward spread across a real-world host landscape, continuing these simulations until a predefined threshold prevalence is exceeded. We then couple these simulation outputs to a stochastic optimisation algorithm designed to select those surveillance sites that maximise the probability of detection of the pathogen, allowing for a range of logistical parameters, such as different sampling intensities and detection abilities. We use this optimisation method approach to answer the question of where in a landscape surveillance should be targeted if we are to maximise the probability of detecting new pathogen incursions before the threshold prevalence is reached. We interpret our results by focussing on the following questions:

1. How much of an increase in detection probability can our method achieve compared to conventional site-selection approaches?

2. How do the locations and frequency of pathogen introductions influence the optimal arrangement of surveillance locations?

3. What is the impact of the number of survey sites, the frequency of surveys, and the diagnostic sensitivity of the detection method on the optimal arrangement of surveillance locations?

4. Can we identify general rules for the selection of surveillance sites that approximate those of optimised surveillance schemes, which thus could be readily deployed in practice without the need for optimisation?

To demonstrate our method in the context of a pressing example, we use HLB (also known as citrus greening)—a high-profile, devastating disease of citrus trees—in the US state of Florida as a case study. HLB is caused by the bacterium *Candidatus* Liberibacter asiaticus (Las) and spread by the Asian citrus psyllid, *Diaphorina citri*, which has been established in Florida as an invasive species since at least 1998 [38]. HLB is currently endemic in the state, where it decimated the citrus industry in less than a decade following first detection in 2005 [39]. We consider here a scenario prior to this incursion, in which the psyllid is present but Las is absent from the state, but in which there is an immediate and ongoing risk of introduction of Las through human movements from other currently infected areas (such as Brazil and China). We use different estimates of where and how often the pathogen is introduced to the state in order to capture the inherent uncertainty in these processes and investigate how these influence the optimal surveillance strategy.

## Materials and methods

### Summary

We consider here how best to deploy surveillance resources in order to maximise the probability of detection before a specified 'maximum acceptable prevalence' is reached. To do this, we developed a grid-based, stochastic, spatially explicit, landscape-scale model and repeatedly simulated pathogen spread through this landscape until this prevalence was reached. Although the model is pathogen-generic, we parametrised it to replicate early stage spread of Las in Florida. We then used an optimisation routine (simulated annealing) in order to identify which arrangement of a specified number of sites, with a fixed number of samples collected per site, would give the highest mean probability of detection over all simulation model realisations, given a particular frequency of sampling using a detection method with known performance

characteristics (the probability of correctly identifying infected hosts—the diagnostic sensitivity—and growth in detectability over time).

## Simulation model

The simulation model is described in more detail in S1 Text. The model runs on a gridded landscape of 1 km × 1 km cells, each containing a density of citrus informed by maps of commercial citrus densities (provided by the US Animal and Plant Health Inspection Service [USDA-APHIS]) and estimates of residential citrus densities (calculated from population data at the census tract level obtained from the US Census Bureau [40]). Individual cells transition stochastically from a susceptible to an infected status in continuous time, driven by pathogen spread from outside the landscape as well as secondary spread within the landscape. Secondary (between-cell) spread occurs according to an exponential dispersal kernel, fitted to data as described in S1 Text. Following first infection, each cell becomes more infectious (and detectable) as infection bulks up locally, again at a rate parametrised using available data. This increase in infectiousness and detectability following detection is deterministic, allowing us to estimate the proportion of both infected and symptomatic plants in each cell at any point in time (given the timing of infection in each cell is known, which will vary for each simulation run). By running this stochastic model a large number of times, we are able to capture the inherent variability in spatiotemporal spread and explicitly account for this when considering the optimal arrangement of surveillance sites.

## Optimisation approach

We consider a set $\Omega$ of $N$ 1-km square grid cells, from within each of which $n$ hosts are assessed (using a detection method with given performance characteristics) every $\Delta t$ units of time. We aim to identify which $\Omega$ (that is, arrangement of surveillance sites) gives the highest probability of detection ($p(\Omega, n, \Delta t)$) before the state-wide prevalence threshold is reached. In the current study, we allowed each site to be selected only once. Because complete enumeration of all possible arrangements is not feasible for a problem of this scale, we used a stochastic optimisation algorithm, simulated annealing, with an exponential cooling schedule [41] to approximate the optimal arrangement of sites, using the output of the simulation model. More details of the algorithm are provided in S1 Text.

## Calculating the probability to detect at least one case of the epidemic

We assume the proportion of detectable hosts in any site increases logistically following first infection, meaning—in site $L$ at time $t$ in model iteration $i$—the proportion of detectable hosts is given by

$$\varphi(L, i, t) = \begin{cases} \dfrac{1}{1 + \left(\dfrac{1}{\varsigma_0} - 1\right)e^{-s(t - \tau(L,i))}} & \text{if } t \geq \tau(L, i) \\[2em] 0 & \text{otherwise} \end{cases},$$

in which $\tau(L, i)$ is the iteration-specific time at which site $L$ first becomes infected, $\varsigma_0$ is the proportion of detectable sites at the time of first infection, and $s$ is the rate of increase in detectability.

This allows us to quantify the probability of failing to detect infection in site $L$ at time $t$ as

$$\varphi(L, i, t, n) = (1 - \eta\varphi(L, i, t))^n,$$

in which each of the $n$ samples has a fixed probability of correctly identifying a detectable host ($\eta$).

The probability of not detecting across the entire sampling pattern is therefore given by

$$\varphi(\Omega, i, t, n) = \prod_{L \in \Omega} \varphi(L, i, t, n).$$

The probability of not detecting in iteration $i$ is therefore given by

$$\varphi(\Omega, i, n, \Delta t) = \frac{1}{\Delta t} \int_{t_0=0}^{t_0=\Delta t} \prod_j \varphi(\Omega, i, t_0 + j\Delta t, n),$$

in which the averaging done by the outer integral accounts for uncertainty in the start of sampling ($t_0$) relative to the time of first introduction of the pathogen anywhere in the landscape ($t = 0$), and the inner product runs over all values of $j$ until the simulation-specific prevalence threshold has been exceeded.

In simulation $i$, the probability of detecting infection given a particular spatial arrangement ($\Omega$), timing ($\Delta t$), and local intensity of sampling ($n$) is the complement of this probability,

$$p(\Omega, i, n, \Delta t) = 1 - \varphi(\Omega, i, n, \Delta t).$$

Our final estimate of the effectiveness of any sampling pattern can therefore be obtained by averaging over the $M$ simulation runs we consider,

$$p(\Omega, n, \Delta t) = \frac{1}{M} \sum_{i=1}^{M} p(\Omega, i, n, \Delta t).$$

This was used as the objective function in the optimisation algorithm, which therefore identifies the components of $\Omega$.

## Code availability

All code for running the simulation model and the optimisation is provided at https://github.com/nikcunniffe/SpatialSampling. Because of commercial sensitivities, we are unable to provide the high-resolution citrus density data used in this report, and so instead we provide citrus density data at the county level, based upon the 2018–2019 Florida Citrus Statistics report.

## Running time

The time required to run the simulation model and the optimisation will depend on the nature of the landscape and the patterns of spatial spread. However, on a 3.00-GHz Intel Xeon processor (Santa Clara, CA, USA), running 1,000 realisations of spread through the state of Florida (using the 'baseline model' parametrisation) takes 166 seconds. Running a single optimisation (again using the baseline parametrisation) with 100,000 iterations on this output requires 411 seconds. This gives a total of 577 seconds.

## Application to HLB

Parameter values for the simulation model (Table 1) were selected using a combination of statistical model fitting, iterative parameter adjustment, and published data and are described in more detail in S1 Text. Because of the level of uncertainty in the rate and distribution of pathogen entry into the state (especially resulting from informal, unreported host movements), we ran different scenarios for these parameters. We considered 'low', 'moderate', and 'high' rates of pathogen entry, for each of which we modelled 2 different spatial patterns of entry:

**Table 1. Description of parameters used in the model.**

| Interpretation | Value | Rationale |
|---|---|---|
| **Simulation model** | | |
| Proportion of citrus in cell. | [Spatial grid] | Estimated as the sum of the commercial and residential citrus density. |
| Citrus density threshold for spread simulation. | 0.0025 km$^2$ | Approximately 1/16th of a typical Floridian citrus grove. Also, this density is around the suspected minimum for citrus canker spread in residential trees in Miami [58]. |
| Maximal overall rate of pathogen entry into the state. | 0.05/year (around one entry every 200 years) | Details on the modelling of pathogen entry are given in S1 Text. These rates were selected in order to represent a variety of different situations. The inverse of each rate relates to the mean number of entries per year if all cells were full of susceptible hosts. Estimates of the modelled rate based on the mean citrus density of 0.1 are shown in parentheses. A variety of other rates were considered when evaluating the impact of model misspecification. |
| | 0.50/year (around one entry every 20 years) | |
| | 5.00/year (around one entry every 2 years) | |
| Initial infectivity following cell infection. | 0.006 | Fitted to data on the spread of disease in Devil's Garden, assuming a logistic increase in infectivity. See S1 Text. |
| Relative rate of pathogen entry. | [Spatial grid] | Two different distributions of pathogen entry were considered, as described in the text. |
| Rate of increase in infectivity per cell. | 1.25 infections/infected/year | Fitted to data from Devil's Garden, assuming a logistic increase in infectivity and a 6- to 12-month asymptomatic period. See S1 Text. |
| Secondary spread parameter (can be interpreted as the maximal number of cells infected by a single infected cell). | 1,000/year | Selected by fixing all other parameters and adjusting until the model simulated state-wide spread in 10 years (see S1 Text). |
| Mean pathogen dispersal distance (assuming exponential kernel). | 20 km (in 2 dimensions) | Estimate taken from recent study of pathogen spread [20]. |
| Maximum prevalence for simulation model. | 1% | Selected as representative of a relatively low prevalence that is also able to capture the impact of spatial autocorrelation following introduction. |
| Number of epidemic realisations to run. | 1,000 | |
| **Detection model** | | |
| Initial probability of detection. | 0.006 | Fitted to data from Devil's Garden, assuming a logistic increase in detectability. See S1 Text. |
| Rate of increase in probability of detection. | 1.00 detectable/detectable/year | Fitted to data from Devil's Garden, assuming a logistic increase in detectability. See S1 Text. |
| Number of sites visited each sampling round. | Varied between 1 and 150 | Optimisation was only performed for between 1 and 50 sites. Site selection using risk metrics was performed for between 1 and 150 sites. |
| Number of samples collected per site each sampling round. | 50 | Considered a reasonable intensity of surveillance within a 1-km$^2$ area. |
| Probability of detection if infected host sampled. | 0.5 | Assuming visual inspection. Estimate obtained from a comparison of PCR and visual inspection for detection of infection in Floridian citrus groves [59]. |
| Interval between sampling rounds. | 1/365 years | Daily |
| | 7/365 years | Weekly |
| | 1/12 years | Monthly |
| | 0.25 years | Quarterly |
| | 0.5 years | Biannually |
| | 0.75 years | Every 9 months |
| | 1 year | Annually |
| **Optimisation** | | |
| Initial temperature. | 10 | Identified by adjusting parameter estimates and inspecting the progression of the objective function (S7 Fig). |
| Rate of cooling. | 0.999 | |
| Number of iterations of simulated annealing algorithm. | 100,000 | |

1. A fixed ('flat') rate, meaning that citrus density alone determined the relative rate of pathogen entry.

2. Variation in rate according to a probabilistic model of likely entry sites (the 'travel census model'). This model describes the relative risk of introduction of Las into each census tract of Florida by accounting for the movement of people into the state from other parts of the world where HLB is endemic [23]. The model predictions were based on data from 2010.

When parametrising the detection process, we considered a 'baseline' scenario in which detection was through visual inspection (as is standard for most plant pathogens [42]) of 50 trees in 20 sites annually (that is, 1,000 trees in total per year), as shown in Table 1. The optimisation algorithm itself was parametrised by applying this baseline detection model to a simulation model parametrised using the values in Table 1 with a low pathogen entry rate, distributed according to the travel census model. We then varied the initial temperature and the cooling parameters of the algorithm and ran the optimisation for 100,000 iterations and inspected the trace of the detection probability. We found little impact of varying these parameters on the objective function of the optimised solution, although the trace plots differed (see S7 Fig). The optimisation algorithm parameters used are shown in Table 1.

### How is the optimal surveillance strategy influenced by risk, detection methods, and epidemiology?

In order to identify the impact of epidemiological and surveillance system characteristics on site selection, we used the following 2 methods to characterise the optimal sites:

- Estimating the mean detection probability. This was performed as described above, using a data set of 1,000 model simulation outputs distinct from those to which the optimisation was applied.

- Visualising and quantifying the spatial arrangement of selected sites by identifying 'clusters' of selected sites. We used a single-linkage agglomerative clustering method to group sites within 20 km of each other (this distance was selected as representative of the mean annual dispersal distance of HLB in Florida [20]). The distribution of these clusters assists in the visualisation of the general arrangement of selected sites, and the total number of clusters provides a useful summary statistic (with lower values indicating more clustering).

We evaluated how well more conventional targeted surveillance approaches perform in comparison to the optimal sites by creating 'risk metrics' accounting for a range of different levels of knowledge about the entry, establishment, and spread of Las (described in the Results). These metrics were allocated to individual sites and the specified number of sites selected (without replacement) with probability proportional to the metric. We repeated this process 100 times for each metric and recorded the detection probability for each run using the data set of 1,000 runs used to estimate the detection probability for the optimal sites.

### Results

Although our spread model identifies a number of areas of high risk of pathogen presence (predominantly centred around areas of high citrus density; Fig 1), when we apply the optimisation routine, we find that the common practice of focussing surveillance in a small number of highest-risk areas generally does not maximise the mean probability of detecting the pathogen (Fig 2). However, we show that the effectiveness of the detection method used to find disease determines how that method should be deployed across the landscape, with poorer

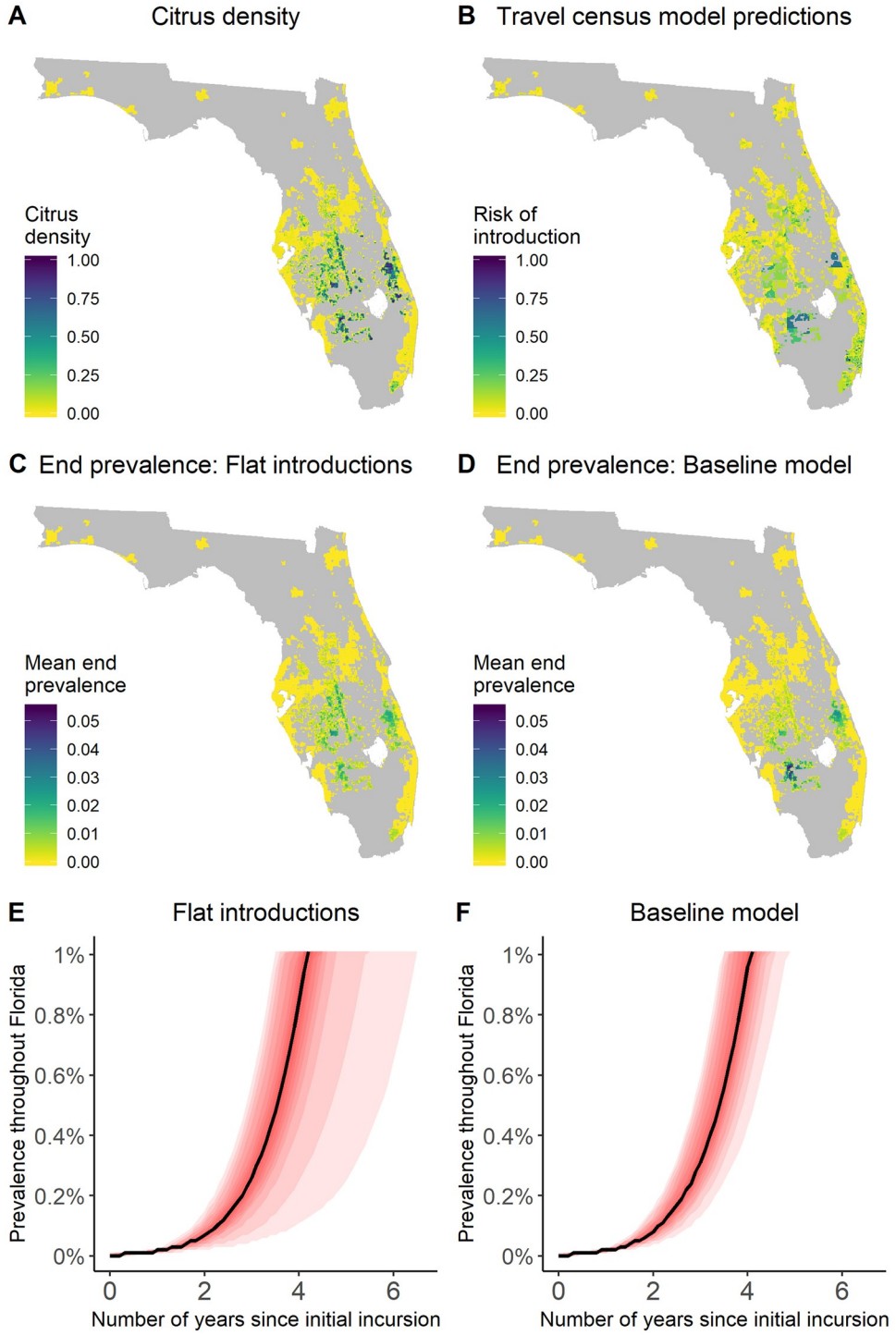

**Fig 1. Simulation model upon which the optimisation is based.** Plot A shows the distribution of citrus trees (which also represents the relative rate of introduction under the assumption of a 'flat' distribution of pathogen entry). Plot B shows the distribution of the relative risk of introduction according to the 'travel census' model (which is combined with the citrus density to estimate the relative distribution of introductions under the 'variable' model [that is, the 'baseline' model]). Plots C and D show the mean end prevalence if introductions are based only on citrus density ('flat' pathogen entry, C) or both citrus density and travel census risk ('variable' pathogen entry, D). Plots E and F show the 5th–95th percentiles of the disease progression curves under each of these assumptions, with greater intensity of colouration for percentiles approaching the median (shown as a solid line). The data used to create these plots can be found at https://doi.org/10.17866/rd.salford.12759929.v1 (files 'spatialData_baselines.csv', 'dpcData.csv', and 'dpcSummary.csv').

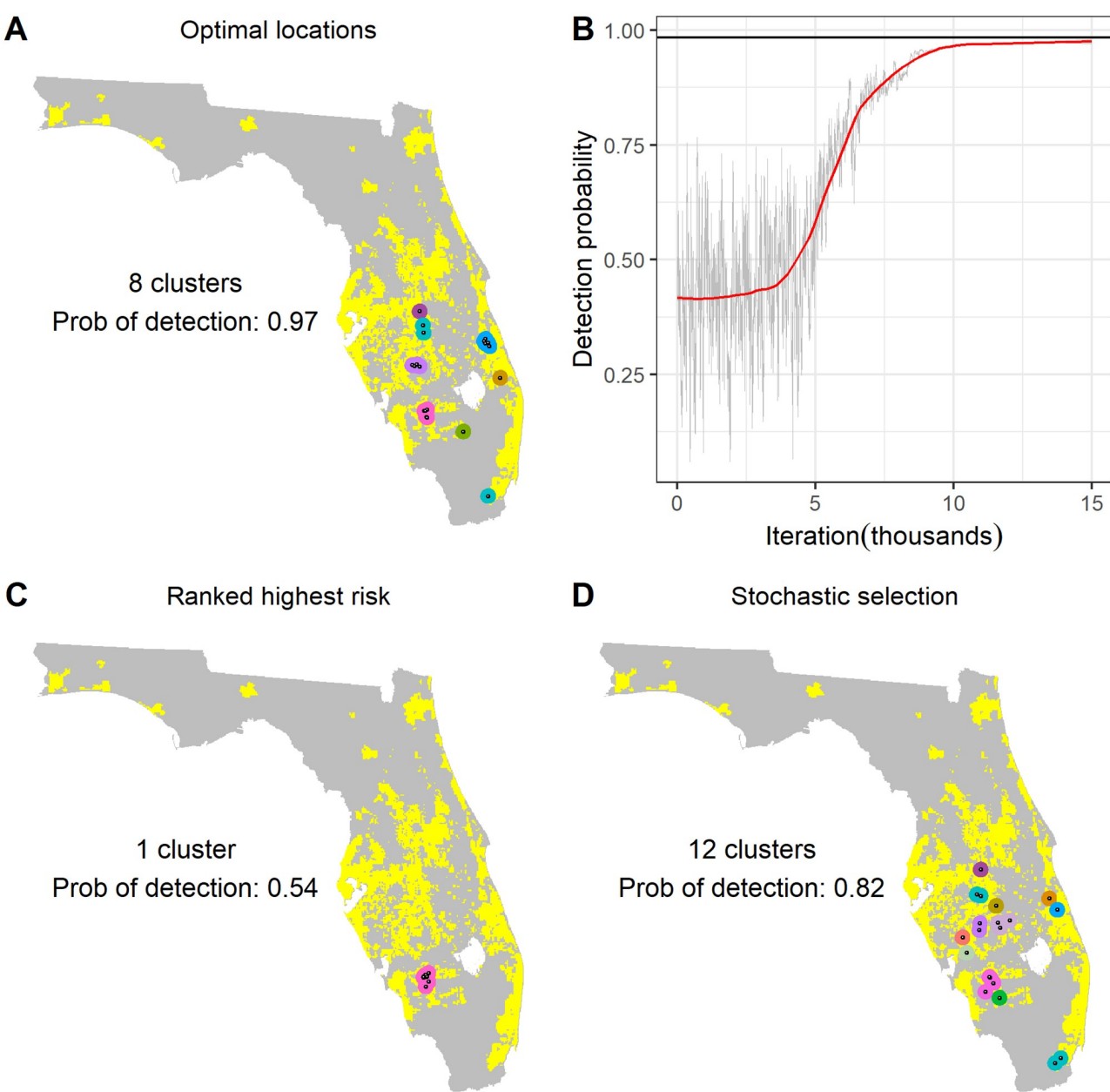

**Fig 2. Arrangement of sampling sites under different selection schemes.** These plots show the distribution of selected sites for 1 case of optimised selection and 2 alternative risk-based approaches based upon the mean end prevalence obtained from the simulation model. Plot A shows the sites selected to maximise the probability of detection when using simulated annealing. The progression of the detection probability over the first 15,000 iterations of the simulated annealing algorithm is shown in plot B, with the solid black line indicating the final detection probability after 100,000 iterations. Plot C shows the 20 sites with the highest mean end prevalence over all realisations, and plot D shows 1 arrangement of 20 sites selected with a probability proportional to the mean end prevalence. Clusters (defined as sites within 20 km of each other) are shown in distinct colours in plots A, C, and D. Estimates of the number of clusters and the probability of detection under the different sampling patterns are also shown. The data used to create these plots can be found at https://doi.org/10.17866/rd.salford.12759929.v1 (files 'spatialData_baselines.csv' and 'ofProgressionExample.csv').

performing methods often requiring a greater focus in a relatively small number of risk 'clusters' (Fig 3).

We compared our method with more conventional targeted surveillance strategies by selecting surveillance sites based on 1 of 4 cell-specific 'risk metrics' that could be expected to

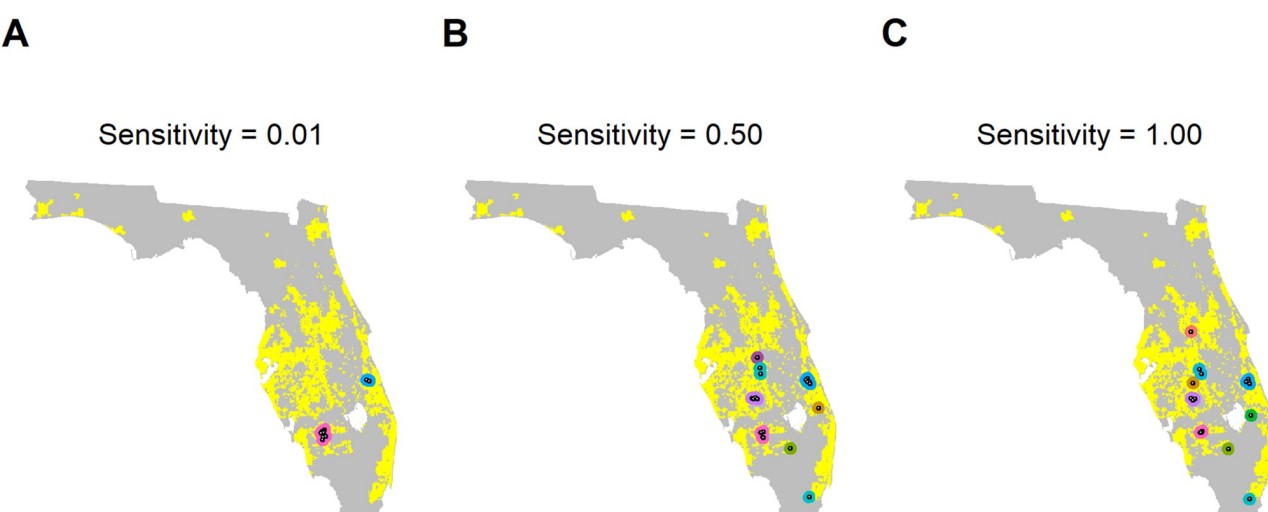

**Fig 3. Impact of test sensitivity on optimal sampling pattern.** These plots show an example of the optimal distribution of sampling sites and clusters (points within 20 km of each other; shown in distinct colours) when the probability of correctly identifying any sampled infected tree (the diagnostic sensitivity) is low (0.01; plot A), medium (0.50; plot B), and high (1.00; plot C). The data used to create these plots can be found at https://doi.org/10. 17866/rd.salford.12759929.v1 (file 'spatialData_baselines.csv').

describe the infection risk for any given site, each representing a different level of knowledge of the likely presence of the pathogen:

1. Random sampling throughout the simulated landscape (indicating no knowledge of risk factors for infection).

2. Relative rate of pathogen incursion, excluding citrus density (indicating known areas of likely pathogen entry).

3. Citrus density (indicating known areas of likely pathogen establishment and spread).

4. Product of citrus density and relative rate of incursion (indicating known areas of likely pathogen entry, establishment, and spread).

We found that for any combination of epidemiological and surveillance parameters, the conventional method of selecting sites based on site-specific 'risk metrics' gives a consistently lower probability of pathogen detection than the optimised approach in almost all cases (even over 100 realisations of these alternative sampling methods; see Figs 4, 5 and 6, S3, S4, S5 and S6 Figs). Our method is also robust to misspecification of model parameters and consistently outperforms alternative methods in these situations (Fig 6). These results clearly demonstrate the importance of carefully considering pathogen epidemiology and entry processes, but also the detection efficiency of inspection and detection technologies, in a holistic manner when planning early detection surveillance.

## Epidemiological simulations

Our model simulates the early stages of an epidemic up to a predefined 'threshold prevalence' of 1%. This particular threshold was arbitrarily selected to indicate a point at which control would no longer be possible in the case of the HLB pathosystem but could alternatively be selected based upon regulatory guidance and/or economic considerations, as described in the Discussion. Since the model is stochastic, we simulate not only the spatial spread within a

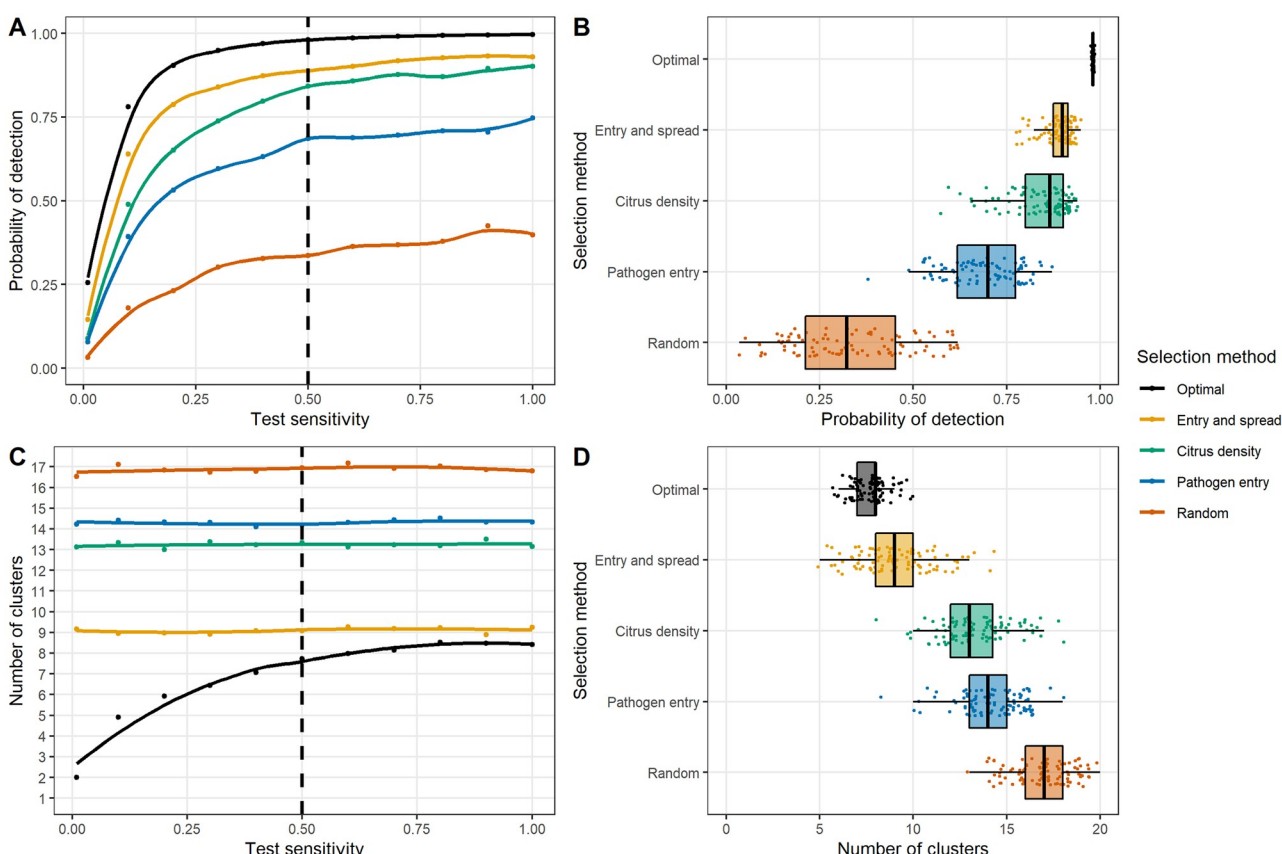

**Fig 4. Impact of varying test sensitivity on detection probability and sampling site clustering.** These plots show the effect of different site-selection strategies on the detection probability or number of clusters. We consider 5 selection strategies: an optimised arrangement and weighted sampling according to 4 different 'risk metrics'—the product of travel census probabilities and citrus density ('Entry and spread'), citrus density, probability of entry according to the travel census model ('Pathogen entry'), and random (that is, unweighted) selection. All selection strategies were repeated 100 times. Plots A and B show the detection probability for these different selection strategies, and plots C and D show the number of clusters (with a cluster being all points within 20 km of each other), all with fitted locally weighted regression curves. Plots A and C show the mean probability of detection or number of clusters, with the vertical dashed line representing the 'baseline' scenario of a diagnostic sensitivity of 0.5. Plots B and D show the variation in individual-level selection runs under this baseline scenario. The data used to create these plots can be found at https://doi.org/10.17866/rd.salford.12759929.v1 (file 'optimisationOutputs_testSens.csv').

landscape over time but how this varies from one epidemic to the next. Taking the average over all of these epidemics, we find that the distribution of entry sites (whether only influenced by the distribution of citrus trees or also by the predictions of the travel census model) affected which sites were more or less likely to be infected by the end of the simulation (Fig 1 and S1 Fig).

## Comparison with conventional targeted surveillance

As a result of spatial autocorrelation (that is, the fact that the risk status of any given site is likely to be similar to that of surrounding sites), we found that simply selecting the sites with the highest risk of infection tended to result in a small number of clusters of sites (Fig 2). To avoid this, we therefore used instead a selection method in which the probability of site selection was proportional to the risk metric, as has been used previously for targeted surveillance [20]. This resulted in a greater spread of selected sites (that is, a larger number of distinct

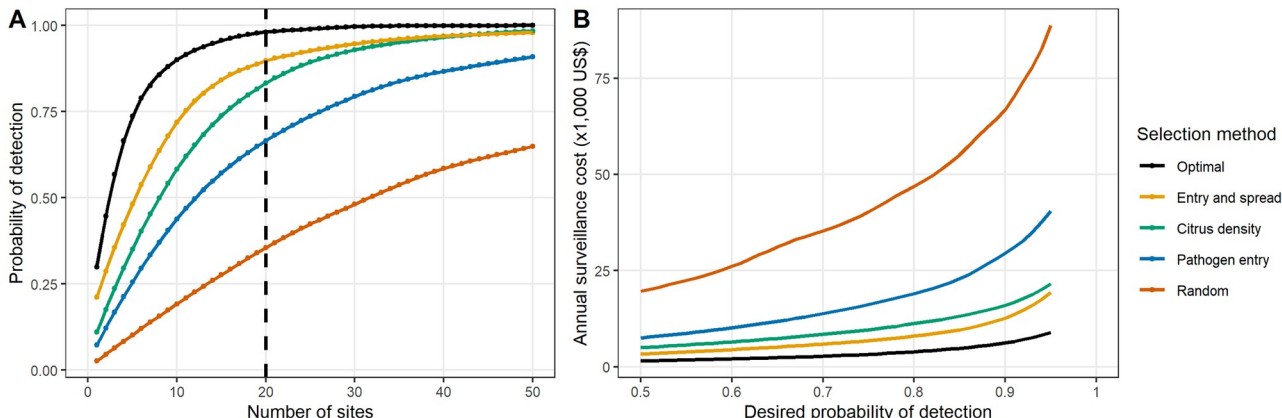

**Fig 5. Impact of varying sample size on detection probability.** These plots show the effect of varying the number of sites—and therefore also the expected cost of surveillance—on the detection probability before the threshold prevalence is reached. Again, we consider a range of selection strategies: an optimised arrangement (based in this case on a single optimisation run for each number of sites) and 100 runs of a weighted sampling strategy based on 4 different 'risk metrics'. These risk metrics are the product of travel census probabilities and citrus density ('Entry and spread'), citrus density, probability of entry according to the travel census model ('Pathogen entry'), and random (that is, unweighted) selection. Estimates of the probability of detection were made for all numbers of sites between 1 and 50 for all selection methods and additionally for all numbers of sites between 51 and 150 for the risk metric strategies, and estimates of the detection probability were interpolated using locally weighted regression. Plot A shows the mean probability of detection for a range of different numbers of sampling locations and demonstrates the variation in probability of detection for any given sample size, with the vertical dashed line representing the 'baseline' scenario of 20 sites. Plot B shows the mean expected annual surveillance costs required to achieve any given probability of detection between 0.50 and 0.95 for the different selection strategies. We assume that the total surveillance cost is the product of the number of sampling sites and the per-site cost of surveillance, as described in the text. The data used to create these plots can be found at https://doi.org/10.17866/rd.salford.12759929.v1 (files 'optimisationOutputs_numSites.csv' and 'costEstimates.csv').

'clusters' of sites) and a higher detection probability than the ranking approach (Figs 2 and 3). The detection probability of optimised sites was consistently higher than that for sites selected using more conventional methods (with the only exception being when the true rate of pathogen entry was underestimated by a factor of 3,200 or more [Fig 6A]). However, of the conventional strategies, the risk metric calculated as the product of citrus density and relative rate of incursion gave the closest detection probabilities to the optimised approach. This was most pronounced in cases for which the rate of pathogen entry was very high (Fig 6). As expected, random sampling gave the lowest mean detection probability, demonstrating the value of targeted surveillance strategies for early detection surveillance.

## Surveillance costs

As expected, for any given set of epidemiological and detection parameters, the detection probability could be increased by increasing the number of sites sampled (Fig 5A). Therefore, in order to achieve any given detection probability between 0.50 and 0.95 using a conventional site-selection strategy, between 2 and 13 times as many sites needed to be sampled as was required under the optimised strategy (S6 Fig). We can also consider this in terms of the total cost of these inspections, as shown in Fig 5B. Here, the mean detection probability was estimated for different numbers of surveillance sites (between 1 and 50 sites for a single run of optimised sites and between 1 and 150 sites for 100 replicates of the other risk metrics). Using rough estimates of sampling costs taken from citrus survey activities undertaken by the United States Department of Agriculture Agricultural Research Service (USDA-ARS) and USDA-APHIS in Florida, a single surveyor would cost around US$120 per day and a single PCR test around US$10 (T. Gottwald, personal communication). Assuming that a single 1-km cell can

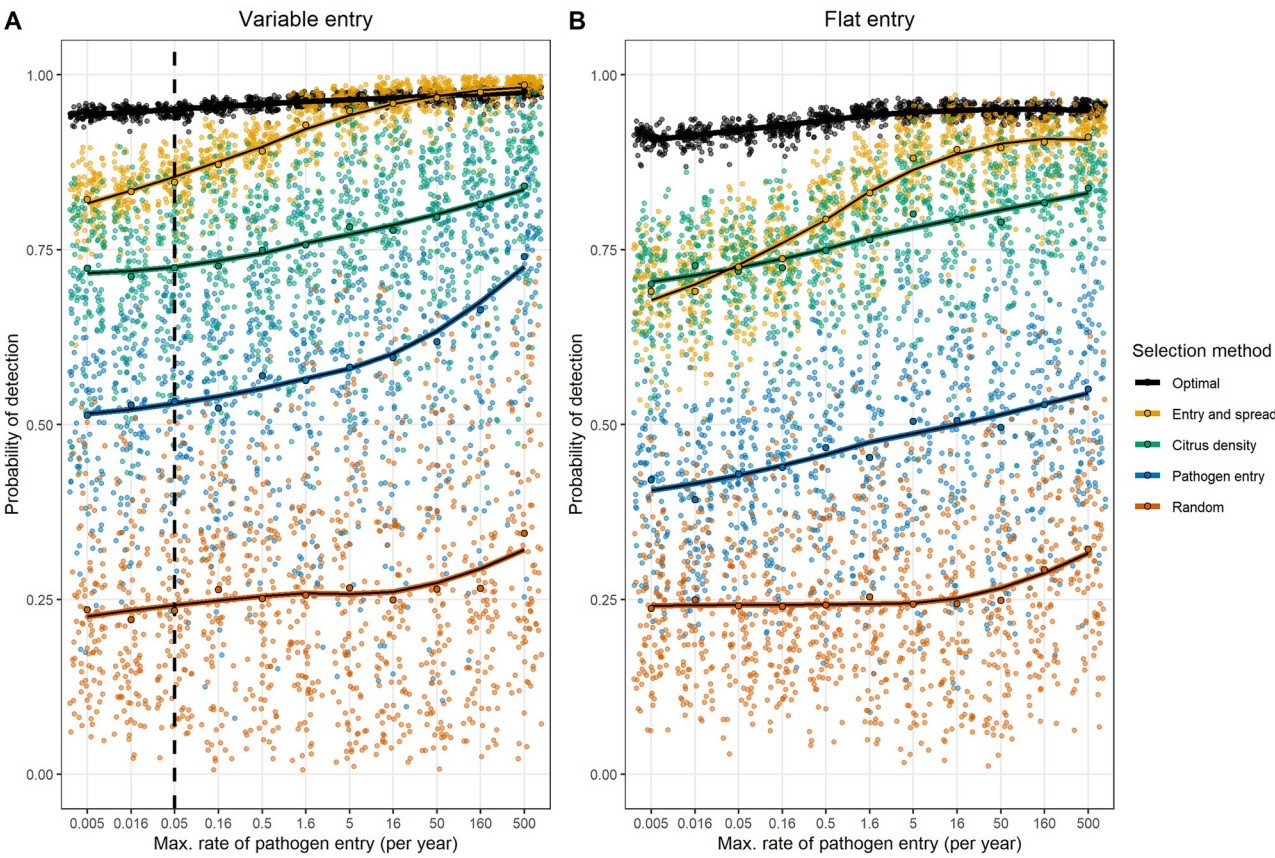

**Fig 6. Impact of incorrect assumptions on performance of different site-selection methods.** These plots show the mean detection probability under different site-selection methods for different rates and patterns of pathogen entry. Plot A shows the detection probability when pathogen entry follows the travel census model, and plot B shows the same when entry is only affected by the citrus density. In all cases, sites were selected under the baseline model assumptions (that is, a maximal rate of pathogen entry equal to 0.05/year and a distribution of entry based upon the travel census model, as shown in the vertical dashed line). Each selection method was repeated 100 times, with each individual detection probability shown as a coloured point. The mean detection probability is shown as a black-bordered point, and a locally weighted regression curve is overlaid to better illustrate the trends. The risk metrics used for conventional targeted selection represent the product of travel census probabilities and citrus density ('Entry and spread'), citrus density, the probability of entry according to the travel census model ('Pathogen entry'), and random selection from the landscape. The data used to create these plots can be found at https://doi.org/10.17866/rd.salford.12759929.v1 (file 'modelMisspecification.csv').

be fully surveyed in a single day by a single inspector and that each of the 50 trees inspected within this cell will be tested with a single PCR test, the total surveillance cost per cell is US $620. This would mean that our 'baseline' surveillance strategy of 20 cells per year would have an annual cost around US$12,400. Fig 5B shows that for any given detection probability between 0.50 and 0.95, the annual surveillance cost using the optimised strategy ranged from US$1,500 to $8,859 (mean: US$3,594). However, the costs of the best-performing conventional strategy ranged from US$3,297 to $19,282, with a mean of US$7,534, and those of random sampling ranged from US$19,606 to $88,689 per year, with a mean of US$42,092 (Fig 5B).

## Varying surveillance characteristics

Under the baseline model (a low rate of pathogen entry, with spatial variation in entry rate, and 20 sites surveyed annually using a detection method with a diagnostic sensitivity of 0.5), the optimised mean detection probability approached 1, with very little variation in the

optimised detection probability between different optimisation runs (Figs 4 and 6, S3 and S5 Figs). There was, therefore, little effect of reducing the sampling interval on the performance of the optimised method (S4 Fig). However, reducing the number of sites visited (Fig 5 and S6 Fig) or the probability of correctly identifying infected hosts (that is, the diagnostic sensitivity of the detection method; Fig 4 and S5 Fig) decreased the mean probability of detection under the optimised strategy. We also found that when the diagnostic sensitivity was high, surveys should be spread throughout the citrus landscape. However, as this was reduced, surveys become increasingly concentrated in a small number of 'hot spots' (Fig 3), where higher rates of pathogen entry intersected with higher citrus density (Fig 1). This was apparent when the numbers of clusters of sites were considered (Fig 4 and S5 Fig).

## Varying the rate and distribution of pathogen entry

Increasing the rate of pathogen entry into the state decreased the number of clusters of surveillance sites (S2 and S3 Figs). This effect was more pronounced when 'high-risk' entry sites were present, reflecting the end prevalence estimates in these sites from the model simulations (S1 Fig). We identified 3 consistent clusters of surveillance locations in citrus growing regions to the northeast, northwest, and southwest of Lake Okeechobee, as well as clusters to the east of the lake and in the centre of the peninsula. The detection probability increased slightly as the rate of pathogen entry increased and was higher when the distribution of introduction points was variable than when it was flat (S2 and S3 Figs), likely reflecting the higher site-specific prevalences (and therefore higher achievable detection probabilities) in these situations. We found that misspecifying the rate of pathogen entry (that is, assuming that the baseline model was correct when the true entry rate was higher or lower than this) had a relatively small impact on the performance of the optimised method, which consistently outperformed all other selection strategies in the vast majority of cases (Fig 6). In all cases, higher rates of introduction were associated with equal or higher detection probabilities (even for optimised selection when site selection was based upon an incorrect model) (Fig 6).

## Discussion

### Model-informed surveillance

In the current report, we describe a novel, to our knowledge, method of identifying how best to deploy surveillance efforts in order to detect epidemics of exotic pathogens at an early stage. We also explore how certain epidemiological and surveillance characteristics impact on the optimal surveillance strategies. Our method links the output of an epidemiologically informed, spatially explicit simulation model capable of reproducing early-stage pathogen spread with an optimisation routine informed by specified surveillance parameters [43]. Our key finding is that it is generally best to avoid 'putting all your eggs in one basket' when planning surveillance and that surveillance resources should generally be spread throughout the landscape to cover all areas of risk (Figs 2 and 3). This is an important message because many surveillance programs in plant health are typically disproportionately targeted to a small number of high-risk areas, such as areas immediately adjacent to current outbreaks or surrounding ports of entry. Suboptimal deployment of surveillance resources such as this can be ill-afforded at a time when the number of plant pests and pathogen threats is rising.

Our method also goes beyond the simple maxim of not putting all eggs in one basket by suggesting precisely which number of baskets should be used and where they should be. We find that the answer to these questions is particularly affected by the performance of the detection method being used (Fig 3), whereas the rate and distribution of pathogen entry is less important (S2 Fig). Although the exact selected sites varied slightly when the optimisation

algorithm was run repeatedly on the same model output, the general locations and arrangement were very similar and resulted in minimal perceivable change in the final detection probability (Figs 4 and 6, S3 and S5 Figs), indicating that the performance of the optimised strategy was robust. We also found that that there was relatively little impact of making the wrong assumption regarding the rate and distribution of pathogen entry on the detection probability of the optimised strategy (Fig 6). Our method considers a surveillance strategy in which a static selection of sites are repeatedly visited, thus bearing similarities to 'sentinel surveillance' [44]. This allows us to better explore the general drivers behind surveillance site placement and how these are impacted by different epidemiological and practical issues. We do not consider an 'adaptive' surveillance strategy, in which surveillance sites are adjusted in light of previous findings. However, using optimisation methods to explore this would potentially have considerable practical value and would be worthy of further attention.

## Optimisation outperforms conventional targeted surveillance strategies because it considers the system as a whole

For all scenarios assessed (including those based on an incorrectly parametrised model), the optimised surveillance strategy outperformed all strategies based on risk metrics (Figs 2, 4, 5 and 6, S3, S4, S5 and S6 Figs). This results from the optimisation being able to explicitly capture relevant aspects of the transmission and detection process and evaluate the surveillance plan whilst considering the system as a whole, thereby identifying and accounting for 'clusters of risk'. This is shown schematically in Fig 7, in which the risk status of the 2 upper locations is correlated (that is, when one site is infected, the other is also likely to be infected). In the presence of an effective detection method, visiting either of these locations would give valuable information on the other, thereby effectively freeing up surveillance resources to be placed in the third site, even though the probability of infection in this site is lower. This complex interplay between the spread of the pathogen and the performance of the detection method requires the consideration of patterns of spread on a run-by-run basis, which lends itself naturally to simulation modelling, which can replicate large numbers of simulation realisations. These patterns cannot be explicitly captured using conventional targeted surveillance strategies based on risk metrics, which are commonly more focused on 'average' patterns of spread, nor by other heuristic methods of site selection [45]. The use of an optimisation algorithm provides a valuable method of collating and interpreting these individual simulation model outputs.

Although it was generally not possible to replicate the performance of the optimal strategy using the conventional targeted surveillance strategy of selecting sites according to site-specific risk metrics, we found that the product of the relative probability of introduction and the citrus density (which effectively describes the expected relative rate of pathogen entry in the model) gave the highest detection probability of the risk metrics we considered (Figs 4, 5 and 6, S3, S4, S5 and S6 Figs). This metric gave a clear improvement in comparison to random sampling or selection based upon relative rate of pathogen entry alone, likely reflecting the ability of this metric to capture both the probability of pathogen entry and its onwards spread (which would be expected to be associated with citrus density). This reinforces the importance of considering pathogen epidemiology when selecting surveillance sites and shares characteristics with our previous targeted surveillance strategy of quantifying risk as the product of the introduction probability and the magnitude of onward spread if introduction occurs [20]. We found that the performance of this risk metric was particularly high when the true rate of pathogen entry was very high and either matched or outperformed the (incorrectly specified) optimised strategy in these cases (Fig 6).

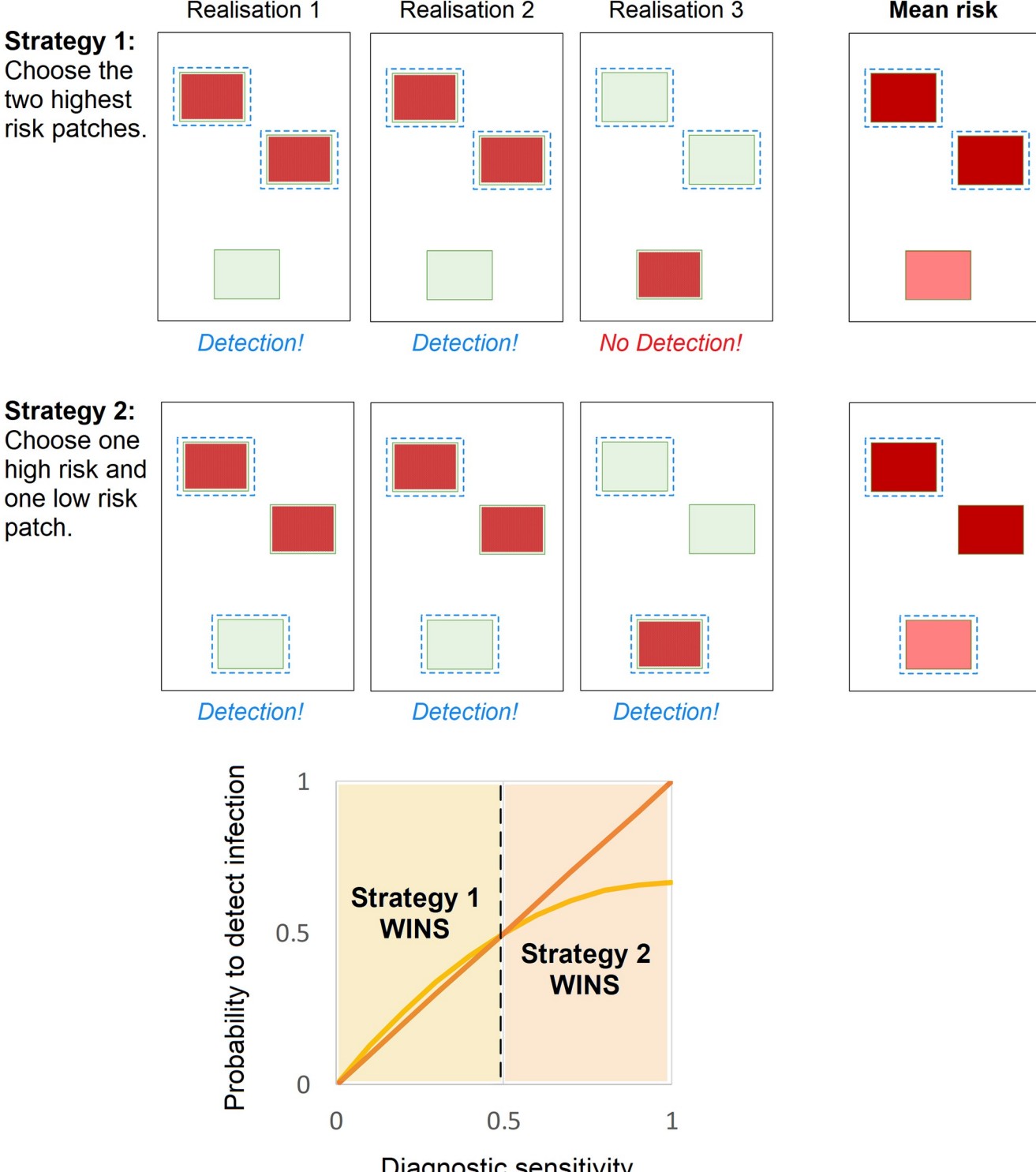

**Fig 7. Maximising the probability to detect a pathogen is achieved not by selecting the highest-risk patches, but by spreading surveillance across all clusters of risk.** For ease of communication, we consider patch status as a dichotomous variable (which can be infected, shown in red, or uninfected, shown in green), rather than considering dynamic trends over time. We wish to sample the 2 patches that maximise the probability of detecting infection over all realisations (selected patches are shown with a blue dashed outline). The 2 'strategy' diagrams on the top each show 3 possible realisations of patch infection status, with the status of the upper 2 host patches correlated due to their proximity. We consider 2 selection strategies: one in which patches are selected based on their mean risk (Strategy 1) and one in which only one high-risk patch is selected and the remaining resources placed in the low risk patch (Strategy 2). If our detection method is perfect, we demonstrate that Strategy 2 outperforms Strategy 1 (being able to detect infection in each of the 3 realisations). The plot on the bottom shows how this is affected by the ability to detect infection in the patch (that is, the diagnostic sensitivity), with the detection probability under Strategy 1 shown in orange and that under Strategy 2 shown in red, for all sensitivity estimates between 0 and 1. In this particular example, the mean

probability to detect under Strategy 1 is calculated as $2 * (1 - (1 - \text{sensitivity})^2)/3$, whereas for Strategy 2 it is equal to the sensitivity. When the sensitivity is low, selection of a single high-risk patch is insufficient to reliably detect infection, and since infection is more common in the uppermost sites, the optimal strategy is therefore to place all the resources amongst these sites (that is, Strategy 1). However, this strategy will never detect the infection in Realisation 3. This limitation becomes more apparent as the diagnostic sensitivity is increased, and beyond a sensitivity of 0.5, Strategy 2 outperforms Strategy 1, with the difference in performance increasing as the sensitivity is increased.

## The performance of the detection method should be considered when planning where to conduct surveillance

An important component of our optimisation algorithm is the ability to explicitly account for parameters which influence the detection ability, but do not affect the epidemiological dynamics of the pest or pathogen itself. Our analysis shows that the performance of the detection method (that is, the diagnostic sensitivity) has a considerable impact on the optimal arrangement of surveillance resources, as well as on the overall detection probability. This finding is due to spatial autocorrelation in the status of individual sites, meaning that the status of each individual surveillance site is not independent of that of nearby sites. As demonstrated in Fig 7, each site inspected provides some information on the status of nearby sites, but the amount of this information decreases as the diagnostic sensitivity of the detection method and/or the number of samples taken per site decreases (and the probability of detecting individual infected hosts decreases). This means that the optimal surveillance strategy for any given pathogen will differ when a highly sensitive detection method is used compared to when the sensitivity is low, with focused sampling in a smaller number of clusters of sites advisable in the latter case (Fig 3). In cases for which the test sensitivity and the number of hosts inspected taken per site are low, the value of optimising the surveillance strategy is reduced, and conventional targeted surveillance strategies may be more acceptable (Fig 4). Further work will investigate the impact of detection lag periods [15,46] on the optimal deployment of surveillance resources.

## Uncertainty in the rate and pattern of introduction has a relatively small impact on where best to conduct surveillance

Although the ability to explicitly account for the processes of pathogen entry ('primary infection') and onwards spread ('secondary infection') is a particular strength of our method, in some cases there may be considerable uncertainty and/or variability in these processes, making them difficult to parametrise. We therefore investigated how the rate and distribution of pathogen entry impact on the optimal surveillance strategy. Whilst the impact of changing these parameters was relatively low, the optimal surveillance sites tended to be clustered in the sites of highest citrus density for high rates of pathogen entry, with surveillance in lower citrus density areas only being promoted as the rate was decreased (S2 Fig). Despite these differences, the impact of misspecifying the rate and distribution of pathogen entry had a relatively small impact on the overall detection probability, which remained high in all scenarios considered (Fig 6). Interestingly, we also found that the detection probability for all site-selection methods increased as the rate of pathogen entry increased (Fig 6), likely reflecting a greater spread of infected sites due to relatively less spread within the state. Investigation of the impact of changes in secondary spread patterns, as well as spread within and between different groups of hosts, will be considered in future work.

## Conclusions for HLB surveillance

Under the baseline assumptions regarding spread and detection, our method suggests that the highest probability of early detection of HLB prior to 2005 would have been achieved by

focussing surveillance efforts in 8 spatial clusters, mainly located in regions of high commercial citrus density in the centre of the peninsula (Fig 3B). We estimate that this arrangement of surveillance sites would have given a high chance (97%) of detecting incursions before a statewide prevalence of 1% was reached. This suggested arrangement differs from the actual surveillance implemented prior to the first detection of HLB in Florida in 2005, which was focused to the southeast of Lake Okeechobee, around Tampa in the west of the peninsula, and around Orlando [39], none of which are suggested by our method. However, we note that the site of the first detection [39] was close to the southernmost of our predicted sites (Fig 2), and subsequent positive detections to the northeast and southwest of Lake Okeechobee [47] were also identified as surveillance targets using our method (Fig 2). Our approach thus has potential implications for early detection programs in areas where HLB is a threat but has not yet been detected, including citrus production areas in Europe.

## Surveillance aims

We focus our attention in the current report on surveillance for early detection rather than other surveillance aims such as prevalence estimation or spatial delimitation [48] (although our method can be adapted to a variety of surveillance aims by adjusting the objective function). This allows us to concentrate on early stage pathogen spread, when spread dynamics are more predictable and thus easier to model (although we appreciate that there may be uncertainty in parameter estimates in these stages [49]). In doing so, we do not explicitly consider the impact of the disease (or any associated control measures). By instead focussing on the probability of detection before a prespecified prevalence threshold is reached, we are better able to explore the impact of epidemiological and diagnostic parameters on the optimal deployment of surveillance resources for early detection. This allows us to draw valuable insights into surveillance strategies, unencumbered by the influence of other factors such as control costs, and thus bears similarities with studies of how to improve sentinel surveillance strategies in networks [33–37] (which, similarly, generally do not consider costs explicitly). On a practical level, our current approach also fits in well with the concept of 'maximum prevalence thresholds' commonly specified when planning conventional regulatory surveillance for regulated pathogens [50,51], making it valuable in a practical context.

## Capturing costs

Our method considers how best to deploy surveillance in order to maximise the probability of detection of new pathogen incursions. Whilst the detection probability is a valuable metric for evaluating surveillance, it does not itself explicitly capture the costs of surveillance and the costs of disease control at the time of first detection or the benefits of disease control associated with earlier detection (that is, the additional economic impact of disease and the disease management costs avoided). Indeed, our decision to exclude these factors from explicit consideration in our method sets it apart from much of the previous work on optimising and improving surveillance strategies. These studies commonly consider the economics of surveillance and control in unison [52], whether through the linking of optimisation routines with simulation models [27–31,53–55] or through the use of simulation models to explore the impact of different surveillance and control strategies in real-world landscapes [56,57]. The aim of these studies has therefore predominantly focused on identifying the optimal balance of surveillance and control intensity required to minimise the total economic impact of invasive species, which can offer valuable practical insights into how surveillance and control strategies interact. By considering surveillance in isolation of control, our approach offers a different perspective on this important issue and demonstrates how noneconomic factors can influence

surveillance performance. We do not attempt to answer the question of whether surveillance is an economically viable strategy (that is, that the economic costs associated with earlier detection are lower than the costs associated with later detection). Instead, this should be considered when the 'maximum acceptable prevalence' is determined. This could be achieved by fixing the surveillance intensity and estimating, for a range of different maximum acceptable prevalences, the expected costs of surveillance, disease impact, and disease management if detection occurs at or before this prevalence and the costs of disease impact and management in the absence of surveillance. Given that surveillance is economically feasible, the maximum acceptable prevalence could then be selected as the point at which the expected costs of surveillance are lowest.

By considering surveillance effort and the performance of the detection method used, our method does lend itself well to the capturing of the costs of surveillance itself in isolation from the costs of disease or control, which can be achieved by simply comparing the number of samples that need to be collected in order to achieve a given probability of detection. This is shown in Fig 5 and S6 Fig, which clearly demonstrate that at least twice as many samples would be required to achieve any given detection probability when conventional site-selection methods are used in contrast to the optimised approach. If we ignore the costs of travelling between sites (which are likely minimal in comparison to the costs of inspection and testing of hosts) and assume a fixed cost of surveillance per site, the use of conventional site-selection strategies would therefore be expected to at least double the costs of surveillance in comparison to the optimised approach. Based on the cost estimates described above, in order to achieve a mean detection probability of 0.95, the optimised strategy would cost US$8,859 per year, in contrast to the US$19,282 per year required to achieve the same mean detection probability using the best-performing conventional strategy. This therefore represents a potential saving of US$10,423 per year. Assuming in our particular case that the first pathogen entry occurred around 1 year after the start of surveillance and took an average of around 7 years to reach the threshold prevalence, this represents a total cost saving of US$83,384 when using the optimised approach. This is a considerable saving, considering the limited funds generally available for plant health monitoring and surveillance. The bulk of this cost is associated with PCR testing, and whilst savings could be made by only testing suspected cases, this would likely require the use of more highly trained surveyors, which would constitute an additional cost in itself. Further exploration of the optimal balance of these costs would be an interesting and valuable area for further exploration.

## Further work

Although the current study is intended to explore optimal surveillance deployment rather than make concrete suggestions for implementation, further work will apply our method to current ongoing threats, such as the spread of Las in California [8] and *X. fastidiosa* in Italy [10]. Further work will also consider the impact of different surveillance aims (such as maximising the number of detections rather than the probability of at least 1 detection) and incorporate disease and control costs and benefits more explicitly within the objective function (by estimating the exact prevalence at first detection and the implications of this for control).

## Supporting information

**S1 Text. Details on the model formulation, optimisation algorithm, and model parametrisation.**
(DOCX)

**S1 Fig. Distribution of mean end prevalence estimates under different patterns of pathogen entry, demonstrating the increase in variability when pathogen entry is more variable.** These plots demonstrate the impact of varying the characteristics of pathogen entry into the state on the mean prevalence at the point the state-wide prevalence threshold of 1% is reached. Plots A–C show the mean prevalence when the probability of pathogen entry into any given site is affected by the density of citrus host and the travel census probabilities, and plots D–F show the mean prevalence when only the citrus density influences the probability of pathogen entry. Plots A and D show a 'low' mean rate of pathogen entry up to 0.05 entries per year, B and E show a 'medium' mean rate of up to 0.5 entries per year, and C and F show a 'high' rate of up to 5 entries per year. Higher rates of entry result in more variability in end prevalence estimates throughout the state. The data used to create these plots can be found at https://doi.org/10.17866/rd.salford.12759929.v1 (file 'spatialData_primaryInf.csv').
(TIF)

**S2 Fig. Impact of rate and distribution of pathogen entry on optimal targeting of surveillance.** These plots show the spatial arrangement of optimal sites (taken from a single optimisation run) and clusters of these sites, along with the detection probability, when the rate of pathogen entry is varied. Plots A–C show the distribution when the probability of pathogen entry into any given site is affected by the density of citrus host and the travel census probabilities. Plots D–F show the distribution when only the citrus density influences the probability of pathogen entry. Plots A and D show a 'low' mean rate of pathogen entry up to 0.05 entries per year, B and E show a 'medium' mean rate of up to 0.5 entries per year, and C and F show a 'high' rate of up to 5 entries per year. Estimates of the number of clusters and the probability of detection under the different sampling patterns are also shown. The data used to create these plots can be found at https://doi.org/10.17866/rd.salford.12759929.v1 (file 'spatialData_primaryInf.csv').
(TIF)

**S3 Fig. Impact of varying rate of pathogen entry on detection probability and numbers of clusters.** These plots show the impact of rate of pathogen entry on the detection probability and number of clusters of sites within 20 km of each other for the different site-selection strategies explored in the manuscript. Plots A and B show the impact of varying the rate of pathogen entry on the overall mean probability of detection (A) and the total number of clusters (B) when the probability of pathogen entry into any given site is affected by the density of citrus host and the travel census probabilities. Plots C and D show the impact of varying the rate of pathogen entry on the overall mean probability of detection (C) and the total number of clusters (D) when only the citrus density influences the probability of pathogen entry. All estimates are taken from 100 realisations for both the optimised sites and sites selected using different risk metrics. These metrics are the product of travel census probabilities and citrus density ('Entry and spread'), citrus density, probability of entry according to the travel census model ('Pathogen entry'), and random selection from the landscape. The data used to create these plots can be found at https://doi.org/10.17866/rd.salford.12759929.v1 (file 'optimisationOutputs_primaryInf.csv').
(TIF)

**S4 Fig. Impact of varying the sampling interval on the detection probability.** This plot shows the detection probability when the interval between sampling rounds is varied for both the optimised approach and a selection of different conventional targeted approaches using a variety of different metrics. All nonoptimised estimates are the mean of 1,000 sampling realisations in which the probability of site selection was based upon the site-specific measure of

interest. These measures are the product of travel census probabilities and citrus density ('Entry and spread'), citrus density, probability of entry according to the travel census model ('Pathogen entry'), or random selection from the landscape. The optimised estimates are the mean of 10 optimisation runs. The dashed line indicates the 'baseline' scenario considered in the manuscript. The data used to create these plots can be found at https://doi.org/10.17866/rd.salford.12759929.v1 (file 'samplingInterval.csv').
(TIF)

**S5 Fig. Impact of varying test sensitivity on detection probability and numbers of clusters.** These plots show the effect of varying the test sensitivity and different site-selection strategies on the detection probability and number of clusters of selected sites. Each of 100 individual site selection runs are shown per selection strategy, along with the locally weighted regression curve. Plot A shows the impact on the probability of detection, and plot B shows the impact on the total number of clusters (with a cluster being all points within 20 km of each other). As well as the optimised strategy, sites were selected using 4 site-specific risk metrics: the product of travel census probabilities and citrus density ('Entry and spread'), citrus density, probability of entry according to the travel census model ('Pathogen entry'), and random selection from the landscape. The data used to create these plots can be found at https://doi.org/10.17866/rd.salford.12759929.v1 (file 'optimisationOutputs_testSens.csv').
(TIF)

**S6 Fig. Impact of varying number of sites surveyed.** This plot shows the effect of varying the number of sites on the detection probability under a range of selection strategies. A total of 100 site selection runs were performed for each strategy and a locally weighted regression curve was fit to each. As well as the optimised strategy, sites were selected using 4 site-specific risk metrics: the product of travel census probabilities and citrus density ('Entry and spread'), citrus density, probability of entry according to the travel census model ('Pathogen entry'), and random (that is, unweighted) selection. Estimates of the probability of detection were made for all numbers of sites between 1 and 50 for all selection methods and additionally for all numbers of sites between 51 and 150 for the risk metric strategies. The horizontal dashed line shows a detection probability of 0.95, and the vertical dashed lines shows the mean number of sites required to achieve this detection probability under the different sampling strategies. The data used to create these plots can be found at https://doi.org/10.17866/rd.salford.12759929.v1 (file 'optimisationOutputs_numSites.csv').
(TIF)

**S7 Fig. Plots of the 'trace' of the simulated annealing algorithm for a range of different parameter values.** This plot shows the change the detection probability as the simulated annealing algorithm progresses over the first 15,000 iterations, using the baseline simulation and detection parameters. The final selected combination of initial temperature and cooling parameters were 10 and 0.9990, respectively (plot in the second column of the third row). The data used to create these plots can be found at https://doi.org/10.17866/rd.salford.12759929.v1 (file 'ofPlots.csv').
(TIF)

## Author Contributions

**Conceptualization:** Alexander J. Mastin, Timothy R. Gottwald, Frank van den Bosch, Nik J. Cunniffe, Stephen Parnell.

**Data curation:** Alexander J. Mastin, Timothy R. Gottwald, Nik J. Cunniffe, Stephen Parnell.

**Formal analysis:** Alexander J. Mastin.

**Funding acquisition:** Timothy R. Gottwald.

**Investigation:** Alexander J. Mastin, Frank van den Bosch, Nik J. Cunniffe, Stephen Parnell.

**Methodology:** Alexander J. Mastin, Frank van den Bosch, Nik J. Cunniffe, Stephen Parnell.

**Software:** Nik J. Cunniffe, Stephen Parnell.

**Supervision:** Timothy R. Gottwald, Frank van den Bosch, Nik J. Cunniffe, Stephen Parnell.

**Validation:** Alexander J. Mastin, Nik J. Cunniffe, Stephen Parnell.

**Visualization:** Alexander J. Mastin, Nik J. Cunniffe, Stephen Parnell.

**Writing – original draft:** Alexander J. Mastin.

**Writing – review & editing:** Alexander J. Mastin, Timothy R. Gottwald, Frank van den Bosch, Nik J. Cunniffe, Stephen Parnell.

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
