## [Editor Report · Decision Letter 0]

7 Feb 2020

Dear Dr Mastin, 

Thank you for submitting your manuscript entitled "Optimising risk-based surveillance for early detection of invasive plant pathogens." for consideration as a Research Article by PLOS Biology.

Your manuscript has now been evaluated by the PLOS Biology editorial staff as well as by an academic editor with relevant expertise and I am writing to let you know that we would like to send your submission out for external peer review.

Please re-submit your manuscript within two working days, i.e. by Feb 09 2020 11:59PM.

Kind regards,

Lauren A Richardson, Ph.D

Senior Editor

PLOS Biology

---

## [Decision Letter · Decision Letter 1]

20 Mar 2020

Dear Dr Mastin,

Thank you very much for submitting your manuscript "Optimising risk-based surveillance for early detection of invasive plant pathogens." for consideration as a Research Article at PLOS Biology. Your manuscript has been evaluated by the PLOS Biology editors, an Academic Editor with relevant expertise, and by three independent reviewers.

You'll see that while all three reviewers are broadly positive about your study, they diverge regarding its readiness for publication in our journal. Only reviewer #2 is largely satisfied; reviewer #1 feels that the lack of biological insight drops it below the level required for PLOS Biology, and reviewer #3 is lacking an economic component.

After discussion with the Academic Editor, we think that adding the extra aspect requested by reviewer #3 would strengthen your study significantly, and we would also recommend that you switch the paper to our "Methods and Resources article type, which does not have a formal requirement for novel biological insight, and which probably suits your study better overall.

IMPORTANT:

a) When re-submitting, please select "Methods & Resources" as the article type. No re-formatting is required, though we appreciate reviewer #2's comment about bringing some methodological detail further forward in the paper (and we note that our "methods at the end" structure is not a hard-and-fast rule).

b) Reviewer #2 also requests that you make the data and code for this paper available. We'd like to emphasise that this is PLOS policy, and will be essential for publication.

c) Regarding the request from reviewer #3 that you "add an economics layer," the Academic Editor says "The authors could be given a little more guidance on how to address the economic problem. If the method does indeed maximise the probability of detection before some specified threshold prevalence is reached, then it might be expected to be cost-effective relative to alternative methods. The authors could be asked to address the cost effectiveness question directly."

In light of the reviews (below), we will not be able to accept the current version of the manuscript, but we would welcome re-submission of a much-revised version that takes into account the reviewers' comments. We cannot make any decision about publication until we have seen the revised manuscript and your response to the reviewers' comments. Your revised manuscript is also likely to be sent for further evaluation by the reviewers.

We expect to receive your revised manuscript within 2 months. Given the current Covid-19 crisis, however, we are prepared to show some flexibility in this timeline.

**IMPORTANT - SUBMITTING YOUR REVISION**

*Re-submission Checklist*

*Published Peer Review*

*PLOS Data Policy*

*Blot and Gel Data Policy*

Sincerely,

Roli Roberts

Senior Editor

PLOS Biology

REVIEWERS' COMMENTS:

Reviewer #1:

This is a very interesting paper comparing the benefits of using a spatially explicit simulation model and optimisation to improve surveillance for achieving higher pest detection probabilities. The strongest aspect of this study is that the model/approach is operational, meaning it can give managers specific locations to target, which a lot of previous studies lack. However, I do not think the results are well enough explained or understood from a biological perspective. We can see that the optimisation routine improves detection probabilities quite dramatically, but why? The authors provide plausible explanations but because everything is numeric and related to a case study parameterisation, I'm left feeling that I don't understand why the optimisation is working so well. Just showing the optimisation works well because it is taking into account the whole problem is not enough, as there is a lot of similar case-studies that show this too. I'd expect more insights and a more interesting scientific story than "we developed a model, used an optimisation routine to improve detection, and it did well." Lots of papers tell that sort of story. For example in the abstract, there is no strong specific scientific insights described in the case study. It is a good, interesting paper, with some novel bits, and it is definitely publishable as is, in a more specialist journal. In my opinion though for a journal like PLOS Biology, I expect more biological insights.

Some other comments

Ln 23 a common name or associated disease might be useful for non-specialists here

Please state the biological description of the objective function, e.g. your goal is to select the spatial arrangement and sampling intensity at sites to maximize the probability of detecting the disease. There are a lot of possible objective functions one could choose, so really important to be explicit before getting into the methods at the end of the paper.

Ln 336 - 360: I found this section especially strong. It relates to work on optimal surveillance in the invasive species/pest literature and why it's different from the approach considered here. The fact that this approach is operational is a major advancement over past work. Some of this could be hinted at in the intro too if space allows [lines 42-50 seem like a natural place].

Ln 400 - 426: Flag what you are optimising over, is n fixed and just optimising over Omega? Also this info is not part of the optimisation "algorithm" it is part of the problem formulation. The optimisation algorithm is simulated annealing. So subtitles are a bit misleading.

Ln 374 - 462: Even after reading the full simulation and optimisation description, it is really unclear whether this is being thought of as a dynamic or static optimisation problem. Can the sampler change sites between rounds for example? My inclination is no. There are fixed spots where sampling occurs and the optimization statically evaluates where to sample once, and then continues to repeat sampling there. If my interpretation is incorrect and there is something in the supplement that spells everything out, that bit needs to be moved to the main text. Only selecting your sites at the first time step seems like a bit of a drawback as this can influence where a manager might choose at the start, if they cannot sample a new site later. If I have this wrong, I think a schematic diagram describing the order of events in the sampling and simulation might be useful.

Reviewer #2:

[identifies himself as Samuel Soubeyrand]

The authors propose an original approach for early detection of emerging infectious diseases of plants. This approach is grounded on a spatially explicit model of pathogen entry and spread, a statistical model of detection and a stochastic optimisation algorithm (simulated annealing). It is compared to several alternative surveillance strategies, including risk-based strategies, under several epidemiological and detection scenarios, and is applied in a real context, namely the emergence of huanglongbing in Florida, USA.

This manuscript describes a very good piece of work that deserves to be published.

* The manuscript and the SI are well written even if the PLOS typical structure (intro - results - discussion - M&M) does not facilitate the reading of such a methodological work. I do not request changing the order of the sections, but I think that locating l.197-208 (slightly rephrased) before the initial presentation of the Figures at the beginning of the Results section would be more convenient and ease the understanding of the figures. 

* L.64-67 : This statement is, from my point of view, exaggerated. The authors present a definitely original work. However, anterior studies targeted quite close objectives. See e.g.:

- Colman E, Holme P, Sayama H, Gershenson C (2019) Efficient sentinel surveillance strategies for preventing epidemics on networks. PLOS Computational Biology 15(11): e1007517. 

-  P. Holme (2018). Objective measures for sentinel surveillance in network epidemiology. Phys. Rev. E 98, 022313  

- Herrera, J., R. Srinivasan, J. Brownstein, A. Galvani, and L. A. Meyers (2016). Disease surveillance on complex social networks. 469 PLoS Computational Biology (12), e1004928.

- Martinetti D., Soubeyrand S. (2019). Identifying lookouts for epidemio-surveillance: application to the emergence of Xylella fastidiosa in France. Phytopathology 109: 265-276. 

I suggest to the authors to re-word the sentence by writing that they propose an original approach for addressing the question l.65-67.

* About the number of repetitions for the optimal strategy. In Figures 3, 4, 5 and S2, the number of repetitions ranges from 1 to 10 for the optimal strategy whereas it ranges from 100 to 1000 for the other strategies. I guess performing the optimal strategy is particularly time-consuming and this explains the use of a reduced number of repetitions (by the way, can the authors provide typical computer time required?). However, the ms would be more convincing with larger numbers of repetitions, especially for providing evidence 1) that a low diagnostic sensitivity or a high rate of pathogen entry leads to a few survey clusters, and 2) that the advantage of the optimal strategy is still significant under incorrect assumptions, for which one has only a single repetition. 

* L.165: (C) should be replaced by (D) and (D) by (E), shouldn't they?

* L.234: Figure S2 instead of Figure S3, isn't it?

* L.354-357: [47] uses a network-based formalization indeed, but the nodes of the network actually correspond to the cells of a grid covering a continuous real landscape, as in the authors' work.

Moreover, the following articles dealing with the search of improved surveillance strategies for Sharka with spatially-explicit stochastic epidemic/observation models are closely related to the points discussed in the last paragraph of the discussion and could also be cited:

- Picard C., Soubeyrand S., Jacquot E., Thébaud G. (2019). Analyzing the Influence of Landscape Aggregation on Disease Spread to Improve Management Strategies. Phytopathology, PHYTO-05. 

- Rimbaud L., Dallot S., Bruchou C., Thoyer S., Jacquot E., Soubeyrand S., Thébaud G. (2019). Improving management strategies of plant diseases using sequential sensitivity analyses. Phytopathology 109: 1184-1197. 

* For reproducibility reasons, for allowing the application of the authors' approach to other case studies, but also for allowing methodological work on the comparison of different optimization approaches for early detection, in the vein of the authors' proposal, it would be relevant to provide the data, the code of the simulator and the code of the surveillance strategies on referenced open-access archives. 

Reviewer #3:

A spatially explicit model of disease spread is developed and applied to the economically damaging huanglongbing disease, to identify the arrangement of surveillance sites that maximise the probability of early detection. The authors are interested in understanding whether surveillance resources should be placed around a single high-risk site or spread across other potential introduction sites as well. Mainland Florida is used as the case study application.

Given that the problem addressed in this paper is one of optimal resource allocation, my comments and concerns focus on the absence of any economic variables in the model. This is despite the importance of economic factors being alluded to: e.g. (line 38) "… an effective early detection strategy is complicated by … limited financial and logistical resources."; (lines 33-34) "the 'maximum accepted prevalence' will be impacted by … likely impact, and the availability and feasibility of control measures."; and (line 49-50) "there is a risk that the surveillance strategy may not be optimally targeted; resulting in … excessive costs"; (line 221) "… demonstrating the value of targeted surveillance strategies …".

So, despite discussion about the importance of considering the performance of the detection method and spatial arrangement, we have no information about the economically optimal strategy. There is no indication of the various costs involved in their surveillance strategies - the assumption is therefore an unlimited budget. Pest managers, who regularly decide how to spend limited resources on disease control, would require information on the costs and trade-offs involved in various surveillance strategies in order to make these complicated resource allocation decisions. In essence, to obtain the greatest benefits from investing scarce resources, surveillance design must explicitly balance the cost of surveillance with the costs of allowing invading populations to go undetected longer (Epanchin-Neill et al., 2014). 

In my view, therefore, the model is incomplete -an economics layer needs to be included in the model and the analysis before the article can be considered for publishing. I leave this to the authors to decide on the best way of including the economics. Note that while some of the important literature on early detection that includes both biology and economics is cited, the authors appear to be critical of these papers in their focus on the optimal balance of early detection surveillance and control intensity required to minimise costs (lines 61-63 and 348-353). Others articles worth reviewing are Kean and Stringer (2019); and Epanchin-Neill and Hastings, 2010).

Linked to the above, the authors claim that no previous studies have addressed the question of where surveillance resources should be located to maximise the probability of detecting an invading pathogen before it reaches a certain prevalence. I would suggest this is because in those previous studies the authors have chosen to solve the much more relevant and pressing problem of developing surveillance strategies that balance the cost and benefits involved in particular designs.

Line 33: 'maximum acceptable prevalence': how is this determined, and could it based on balancing the cost of surveillance with the costs of allowing invading populations to go undetected longer? 

Line 78-79: "How easy is it to implement this new strategy" - what does 'easy' mean in this context and how will it be evaluated?

Line 86: what is meant by 'general surveillance'?

Line 189-190: the model uses a pre-defined 'threshold prevalence' of 1% to indicate the point at which control would no longer be possible. This seems like an arbitrary value, and some additional explanation is required. For example, has the value of control at a prevalence of 1% been estimated? What is 'control'?

Table 1: 'pathogen entry', could additional explanation be provided on how entry is 'seeded' - is it a random process in a particular cell?

In summary, adding an economic component to the model would make it a very publishable paper.

References

Bradhurst, R. A., et al. (2015). "A hybrid modeling approach to simulating foot-and-mouth disease outbreaks in Australian livestock." Frontiers in Environmental Science 3(17).

Epanchin-Niell, R. S. and A. Hastings (2010). "Controlling established invaders: integrating economics and spread dynamics to determine optimal management." Ecology Letters 13(4): 528-541.

Epanchin-Niell, R. S., et al. (2014). "Designing cost-efficient surveillance for early detection and control of multiple biological invaders." Ecological Applications 24(6): 1258-1274.

Kean, J. M. and L. D. Stringer (2019). "Optimising the seasonal deployment of surveillance traps for detection of incipient pest invasions." Crop Protection 123: 36-44

---

## [Decision Letter · Decision Letter 2]

23 Jul 2020

Dear Dr Mastin,

Thank you for submitting your revised Research Article entitled "Optimising risk-based surveillance for early detection of invasive plant pathogens." for publication in PLOS Biology. I have now obtained advice from two of the original reviewers and have discussed their comments with the Academic Editor. 

Based on the reviews, we will probably accept this manuscript for publication, assuming that you will modify the manuscript to address the remaining points raised by the reviewers. Please also make sure to address the data and other policy-related requests noted at the end of this email.

IMPORTANT:

a) Please attend to the remaining requests from reviewer #3.

b) Please address my Data Policy requests (see further down).

We expect to receive your revised manuscript within two weeks. Your revisions should address the specific points made by each reviewer. In addition to the remaining revisions and before we will be able to formally accept your manuscript and consider it "in press", we also need to ensure that your article conforms to our guidelines. A member of our team will be in touch shortly with a set of requests. As we can't proceed until these requirements are met, your swift response will help prevent delays to publication.

*Copyediting*

*Published Peer Review History*

*Early Version*

*Submitting Your Revision*

Sincerely,

Roli Roberts

Senior Editor

PLOS Biology

DATA POLICY:

Regardless of the method selected, please ensure that you provide the individual numerical values that underlie the summary data displayed in all main and Suppementary Figure panels as they are essential for readers to assess your analysis and to reproduce it. NOTE: the numerical data provided should include all replicates AND the way in which the plotted mean and errors were derived (it should not present only the mean/average values).

REVIEWERS' COMMENTS:

Reviewer #1:

The authors have mostly addressed all my comments adequately. The abstract is especially more insightful, adding the "correlation" component, I can now better understand why its working. I think the new format is also beneficial. My only issue is that it is not explicitly stated in the optimisation summary whether n is fixed beforehand, or whether sets of sites that vary in size are considered by the optimization routine. I assume the latter, but the text makes it seem like the former, but doesn't explicitly say so.

Reviewer #3:

This paper, as suggested in the abstract (line 5) is about "how best to target surveillance resources to achieve [detection of epidemics at an early stage]". This is therefore a problem about optimal resource allocation, thus the costs and benefits of control must be considered properly. I acknowledge that the discussion of the economics aspects of the problem has improved - e.g. control costs have been included -but the benefits of control (damages/impact avoided because control is applied) are also an important consideration and are not fully discussed. There is a trade-off between spending on surveillance for HLB and the cost of controlling future spread which would otherwise cause damage. This trade-off will also be important in choosing the 'threshold prevalence', but is still absent from the discussion.

When considering threshold prevalence the authors acknowledged that this represents 'the highest prevalence at which control could still be feasible'. The term 'feasible' is key, and again, has economic connotations that can't be ignored. Feasibility includes the costs of control and the benefits of control - the damages that are avoided because of control. Benefits should be incorporated into the discussion.

Other comments have been dealt with appropriately.

---

## [Editor Report · Decision Letter 3]

14 Sep 2020

Dear Dr Mastin,

On behalf of my colleagues and the Academic Editor, Charles Perrings, I am pleased to inform you that we will be delighted to publish your Research Article in PLOS Biology. 

Early Version

PRESS 

Kind regards,

Vita Usova,

Publishing Editor 

PLOS Biology

on behalf of

Roland Roberts,

Senior Editor

PLOS Biology